# lesSDRF is more: maximizing the value of proteomics data through streamlined metadata annotation

Tine Claeys[1,2], Tim Van Den Bossche [1,2], Yasset Perez-Riverol [3], Kris Gevaert[1,2], Juan Antonio Vizcaíno [3] ✉ & Lennart Martens [1,2] ✉

Public proteomics data often lack essential metadata, limiting its potential. To address this, we present lesSDRF, a tool to simplify the process of metadata annotation, thereby ensuring that data leave a lasting, impactful legacy well beyond its initial publication.

The life sciences have clearly highlighted the potential impact of open science, with groundbreaking discoveries (from structures in the Protein Data Bank[1] to the AlphaFold[2] model) being openly shared, enabling global collaboration and advancing biological understanding. In the field of proteomics, public data sharing became general practice with the establishment of the ProteomeXchange (PX) consortium in 2011. PX centralized the main proteomics repositories, standardized data submission and mandated FAIR (Findable, Accessible, Interoperable, Reusable) principles compliance. The resulting widespread access to publicly available proteomics data has not only allowed researchers to reuse, (re-)analyze, (re-)interpret, and integrate raw data from various experiments, yielding new insights and optimizing data potential, but it also plays a pivotal role in enhancing the reproducibility of research findings. A prime illustration of the utility of data sharing is the development of widely used, data-driven tools such as MS²PIP[3], DeepLC[4] and Prosit[5]. However, a critical hurdle that prevents public proteomics data from reaching its full potential is the lack or limited amount of metadata annotation in repositories and research articles[6–9]. Notably, the aforementioned tools were successful because they do not require biological metadata, being based solely on machine-encoded technical aspects. However, as soon as the objectives of a meta-analysis include deeper biological understanding, any lack of biological metadata becomes a major obstacle.

To address this issue, the Sample and Data Relationship Format (SDRF) for Proteomics (SDRF-Proteomics) was introduced in 2021[10]. This tab-delimited format maps data files to sample characteristics. It supports annotation of (i) biological metadata; (ii) the relationship between a sample and its respective data files; (iii) the technical metadata; and (iv) the factor values outlining the studied variables. All these properties are encoded as ontology or controlled vocabulary (CV) terms, thus ensuring a standardized representation[10]. However,

the format's versatility to grasp multiple use cases also brings considerable complexity. Moreover, due to the absence of a streamlined method for annotation, users currently rely on laborious manual annotation using spreadsheet software such as Excel, while varying ontology use leads to poor machine readability and inconsistencies. As a result, only 156 of 9671 datasets (1.6%) submitted since SDRF's introduction in PRIDE in mid-2022 contain submitter-supplied SDRF annotation. However, as SDRF is also used in large-scale, post-hoc annotation initiatives using an existing platform on GitHub[11], with 220 PRIDE projects annotated so far, the total percentage of SDRF annotated projects in PRIDE since mid-2022 is 3.9%, still a low figure.

## Results and discussion

In the context of a large-scale reprocessing effort, we assessed the availability of metadata in 241 PRIDE projects (datasets) and their accompanying research articles, observing a severe lack of comprehensive metadata annotation throughout. While our primary focus was on metadata within PRIDE, it is worth noting that this dearth of metadata annotation is a common trend for all other proteomics repositories within PX. Details on dataset selection are provided in the Supplementary Information. Figure 1 shows major gaps in metadata provision in both research articles and PRIDE metadata. Even when present, metadata was often scattered throughout the research article and rarely structured. Some missing metadata, such as protein or peptide labeling approaches, could substantially hinder future reuse. Although details on technical aspects, such as mass tolerances, can be inferred from the raw data by tools like pride-asap[12], this process is time-consuming and remains error-prone. Comparison of metadata between the article and the corresponding PRIDE project revealed a substantial number of discrepancies, especially regarding tissue annotation. PRIDE often lacks detailed annotations, for example, using

[1]VIB-UGent Center for Medical Biotechnology, VIB, 9000 Ghent, Belgium. [2]Department of Biomolecular Medicine, Ghent University, 9000 Ghent, Belgium. [3]European Molecular Biology Laboratory, European Bioinformatics Institute (EMBL-EBI), Cambridge CB10 1SD, UK. ✉e-mail: juan@ebi.ac.uk; lennart.martens@UGent.be

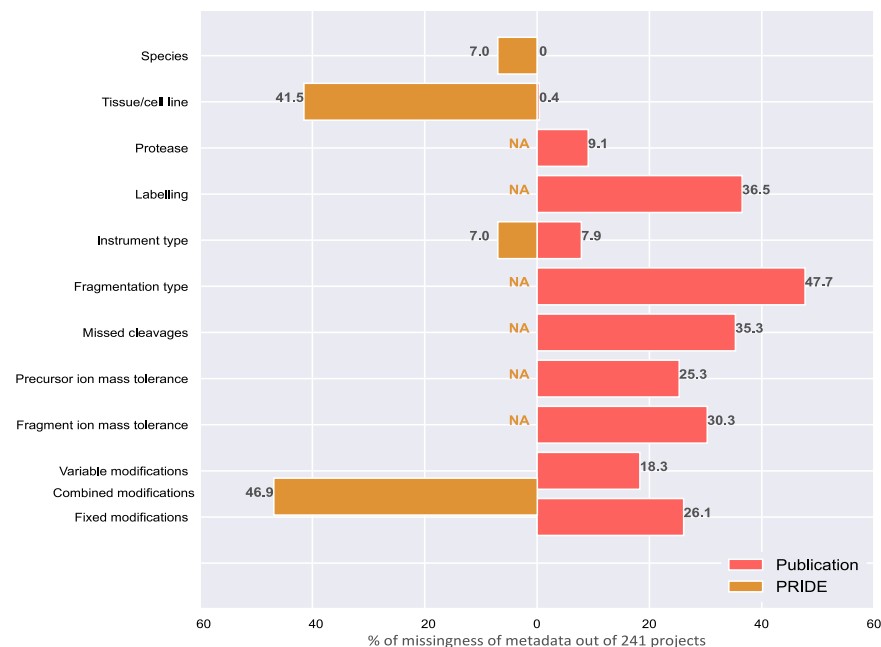

**Fig. 1 | Comparison of metadata incompleteness in research articles and the corresponding PRIDE annotations.** NA indicates that a specific placeholder for this metadata element was not available in the PRIDE structured metadata. To generate this comparison, metadata from both sources was evaluated across 241 PRIDE projects. Bars and numbers indicate the percentage of projects with missing annotation (out of a total of 241 projects). Source data are provided as a Source Data file.

only the broad term "cell culture". The large number of incomplete annotations regarding protein modifications in both sources further indicates that important process-related metadata is often omitted.

To successfully reuse a PRIDE dataset, researchers must thus consult and often interpret repository metadata, the associated paper, its supplementary information, and resort to additional metadata extraction tools, all the while keeping a sharp eye out for inconsistencies or even outright contradictions between these various sources. Clearly, this not only impedes maximal value extraction from public data, especially in large-scale data reuse, but it also raises significant questions about research reproducibility.

Obviously, these issues could easily be avoided if researchers captured metadata prior to submission, preferably in the SDRF standard. We therefore developed a user-friendly, web-accessible application called lesSDRF to streamline annotation of proteomics datasets. The application is structured in five intuitive steps that guide users through the annotation process while integrated ontologies ensure SDRF validity and a seamless user experience. The first step involves species selection, initiating the corresponding SDRF template for raw file name entry. The second step entails uploading a local metadata file, which is then mapped to the SDRF columns while ensuring ontology compliance. In the third step, users can add labeling information (e.g. TMT), which generates new rows based on the label channels. The fourth and fifth steps involve the entry of both required and additional columns using ontology terms which can be selected through a tree-wise representation or an autocomplete search function. To minimize errors, users are allowed to input ontology terms from a drop-down menu. lesSDRF is available at https://lessdrf.streamlit.app/, and a manual is available on the lesSDRF GitHub page (https://github.com/compomics/lesSDRF/blob/main/lesSDRF_manual.pdf).

While lesSDRF simplifies the SDRF-annotation process, some design decisions should be noted. To maintain optimal performance, we set a maximum limit of 250 files that can be processed at once. For larger datasets, users can first generate a partial SDRF, re-upload it into lesSDRF, and modify as needed. Furthermore, it is worth noting that the number of annotation columns within lesSDRF is intentionally limited. We believe this is in the best interest of users as it reduces the

likelihood of confusion by having too many options, while still resulting in a valid SDRF file. Should users require the addition of a new column not available within the app, or that requires a free text input, they can simply download the created SDRF file into a spreadsheet software, such as Excel, and add the required column, following SDRF guidelines.

While lesSDRF aims to simplify and streamline the annotation process, the broader issue of lacking metadata became very clear in our examination, emphasizing the need to improve annotation and dissemination of metadata. Fortunately, there are several ways in which metadata annotation can be improved. These include improvements at the repository level, encouraging detailed dataset descriptions, and measuring dataset impact (e.g., PRIDE datasets in omicsDI[13]) as a powerful incentive for researchers. Moreover, journals can assess metadata annotations prior to publication either manually or using automated tools like *Nature*'s Metadata Creator. But ultimately, the final responsibility for accurate annotation rests first and foremost in our hands, the scientific community. We evolved from (pointlessly) hoarding data to openly sharing these. Now we should do the same for metadata annotation of these data. With lesSDRF we aim to get one step closer to this goal. By integrating feedback of experts from the proteomics community, we ensure lesSDRF remains up-to-date with the evolving field. This is exemplified by the Metaproteomics Initiative's[14] suggestions for metaproteomics-specific SDRF columns that users can import with a click, boosting efficiency. Furthermore, this community effort extends beyond the lesSDRF application itself. In close collaboration with PRIDE, we will integrate lesSDRF into their existing pipeline and annotation efforts. This integration will be twofold. Prior to submission, lesSDRF generated SDRF files will contain a unique hash code recognized by the PRIDE submission system, simplifying the submission process for users. Additionally, for post-submission annotation of existing datasets, we plan to incorporate lesSDRF into the ongoing dataset annotation effort using GitHub, allowing users to upload partially annotated SDRF files into the app. This incorporation of lesSDRF into the PRIDE pipeline does not only mark a significant step towards better metadata management and annotation, but also sets a potent example for other popular tools in

the field. Indeed, we are actively collaborating with prominent proteomics tools such as MSFragger[15], MSstats[16], MSqRob[17] and Mascot[18] to obtain built-in SDRF output, thus also setting a compelling precedent for other tools to follow suit; further increasing all-round reusability of proteomics data.

Through collaboration, we can enhance metadata annotation in proteomics, enabling greater insights and scientific advances from public data. However, no application can fix inaccurate annotation, and submitters thus play a critical role. Indeed, accurate metadata annotation should be considered essential in good scientific practice, as the vast potential of public data reuse is increasingly evident. Accurate metadata provision will therefore ensure that your experimental data will likely become one of your most productive and enduring legacies.

## Methods
### Text mining effort
To extract metadata from public datasets in the PRIDE database, we employed manual text mining using ontology terms obtained from the OLS (Ontology Lookup Service, https://www.ebi.ac.uk/ols/index). A group of seven 3rd year Biotechnology students selected a random sample of projects from PRIDE in the context of a general data reprocessing effort. This resulted in the annotation of 355 proteomics projects along with their accompanying research articles. Of these projects, 262 had associated open-access research articles in PubMed Central that were retained for further analysis. Moreover, we removed 21 protein cross-linking experiments that did not fit with the original objective of the reprocessing study, resulting in 241 projects for further metadata annotation (Supplementary Fig. 1). Metadata categories included biological metadata (species and tissue/cell line) and technical metadata (protease, labeling technique, instrument, fragmentation type, number of missed cleavages, precursor ion mass tolerance, fragment ion mass tolerance, variable modifications, and fixed modifications). Following annotation, all students and supervisors conducted peer reviews to ensure accuracy and consistency of the annotated projects. A list of all the PRIDE dataset identifiers (PXD) of the annotated projects is included as Supplementary Data 1, the results of the text mining were compiled into a final csv file, which is available as Supplementary Data 2 and Supplementary Data 3 contains an overview summary of the different experiment types, labeling methods and sample types identified.

### Development of the lesSDRF web application
The *lesSDRF* web application was built using Streamlit version 1.19.0 and Python version 3.9.13. The following ontologies/CVs were downloaded: PRIDE CV (version 2022-11-17), PSI-MS (version 2022-09-26) and NCBITaxon (version 2022-08-18) in obo format, CL (version 2022-12-25) and HANCESTRO (version 2.6) in OWL format, and EFO (version 3.49.0) in JSON format. Data from the Unimod database for protein modifications was also copied in csv format from their website. Regular updates of these ontologies are scheduled. The downloaded ontologies were stored and parsed into three types of JSON files. First, all the elements from the ontology were stored into a list, which was then stored as an "all_elements.json" file. Second, the ontology was stored as a nested dictionary reflecting the ontology tree. Lastly, the ontology was stored as a tree structure compatible with the streamlit_tree_select module from https://github.com/Schluca/streamlit_tree_select which is used to generate the ontology tree visualization and is referred to as a "nodes.json" file. All JSON files were gzipped to reduce space. The home page of the application used the SDRF templates based on species from https://github.com/bigbio/proteomics-sample-metadata/blob/master/sdrf-proteomics/README.adoc as a starting format of the SDRF. This format consists of all the *required* columns. Additional columns, which were selected based on the same GitHub page and personal experience, can then be added in the next stage.

The editable data frames were generated using the streamlit AG grid module from https://github.com/PablocFonseca/streamlit-aggrid. All code used to create lesSDRF is available via https://github.com/compomics/lesSDRF.

Required packages and their versions are: pronto == 2.5.3, streamlit == 1.19.0, streamlit-aggrid == 0.3.4.post3, streamlit-tree-select == 0.0.5, jsonschema == 4.17.0, zipp == 3.10.0, openpyxl == 3.1.1.

## Data availability
The text mining data generated in this study are provided in the Supplementary Data 2. All used PRIDE projects and their identifiers can be accessed in Supplementary Data 1. The lesSDRF data are freely available on the lesSDRF GitHub page https://github.com/compomics/lesSDRF with https://doi.org/10.5281/zenodo.8406625[19]. Required packages are pronto == 2.5.3, streamlit == 1.19.0, streamlit-aggrid == 0.3.4.post3, streamlit-tree-select == 0.0.5, jsonschema == 4.17.0, zipp == 3.10.0, openpyxl == 3.1.1. Ontologies included are Streamlit version 1.19.0 and Python version 3.9.13 PRIDE CV (version 2022-11-17) PSI-MS (version 2022-09-26) and NCBITaxon (version 2022-08-18) Unimod (version 2023-09-01) in obo format CL (version 2022-12-25) and HANCESTRO (version 2.6) in OWL format and EFO (version 3.49.0) in JSON format. Source data are provided with this paper.

## Code availability
All code is freely available on the lesSDRF GitHub page https://github.com/compomics/lesSDRF with https://doi.org/10.5281/zenodo.8406625[19].

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

## Acknowledgements

We would like to extend our gratitude to the following people who participated in the manual annotation process: Dr. Surya Gupta, Akanksha Negi, Sourav Das, Pragya Srivastava, Diksha Gupta, Anushka Pandey, Amisha Kumari, Manshaa Chaudhary and Dr. Meenu Singh. T.C. received funding from the Research Foundation Flanders (FWO) [1S57123N]. T.V.D.B. acknowledges funding from the Research Foundation Flanders (FWO) [1286824N]. K.G., J.A.V. and L.M. acknowledge funding from the European Union's Horizon 2020 Programme (H2020-INFRAIA-2018-1) [823839]. L.M. acknowledges funding from the Research Foundation Flanders (FWO) [G028821N][G010023N] and from Ghent University Concerted Research Action [BOF21/GOA/033]. J.A.V. and Y.P.R. acknowledge funding from Wellcome Trust [grant number 223745/Z/21/Z] and BBSRC [grant numbers BB/T019670/1 and BB/X001911/1]. J.A.V. and L.M. acknowledge funding from the Research Foundation Flanders (FWO) [W001120N] and from an ELIXIR Implementation study. J.A.V. also acknowledges EMBL core funding.

## Author contributions

T.C. developed the concept, curated and analyzed the data, designed the tool and wrote the paper. J.A.V. and Y.P.R. supported tool development. T.V.D.B., J.A.V., Y.P.R., K.G., L.M. revised the paper. T.C., J.A.V, Y.P.R, T.V.D.B., K.G. and L.M. discussed the results and commented on the manuscript at all stages.

## Competing interests

The authors declare no competing interests.
