## [Peer Review File · Nature Communications]

lesSDRF Is More: Maximizing The Value Of Proteomics Data Through Streamlined Metadata AnnotationREVIEWER COMMENTS

Reviewer #1 (Remarks to the Author):

General comments

This manuscript introduces the work on developing a web-based tool for the Sample and Data Relationship Format (SDRF) file generation. The items of SDRF are collected from the GUI input by the user in a structured way. The author performed a baseline investigation to illustrate the necessity of SDRF file attachment in the submission of public data. The main text of the manuscript was used to introduce the significant meaning of publicly available metadata and the design of lesSDRF, but less attention was paid to the metadata contained and how the metadata can affect the re-use of the data. lesSDRF, as only an orphan web-based tool, may be useful to some academic cases, but we can also find some different tools on the PRIDE official website which can be used in the workflow of data submission. It is more convenient to use the official tools. For another consideration, we cannot figure out that lesSDRF can update automatically as the SDRF metadata standard updating from HUPO PSI. The metadata standard and the knowledge base of lesSDRF were not proven to be authoritative, comprehensive, or useful in practice.

Special Concerns

In the Text mining, the 241 projects included in the statistics were published before 2018, the metadata may not be so concerned in those early submitted projects, these projects submitted recently may be better after the publication from Dai, C. et al. (Dai C et al. A proteomics sample metadata representation for multi-omics integration and big data analysis. Nat Commun. 2021 Oct 6;12(1):5854.). As we know, the number of submitted datasets to PRIDE has increased quickly in recent years.

Ref 23 and 25, why not use a journal-published citation?

Dai C, et al. A proteomics sample metadata representation for multi-omics integration and big data analysis. Nat Commun. 2021 Oct 6;12(1):5854.

Bouwmeester R, Gabriels R, Hulstaert N, Martens L, Degroev S. DeepLC can predict retention times for peptides that carry as-yet unseen modifications. Nat Methods. 2021 Nov;18(11):1363-1369.

Reviewer #2 (Remarks to the Author):

Claeys and colleagues propose an analysis of the annotation of publicly available proteomic data sets. As rightfully stated by the authors, the importance of metadata cannot be understated. They identify a substantial share of missing or incorrect annotation impairing the reuse of these data. To address this issue, the authors propose a web application that allows the generation of standardised metadata annotation. A much welcomed feature is the ability to load and extend partial files. As far as I tested, the tool works as wanted, providing much needed guidance on what the different fields and ontologies refer to. The code is publicly available and a manual is provided along with the tool. Whilst I support the publication of this work, I would appreciate it if the authors could consider the minor suggestions listed in order of appearance in the text.

1- "the life sciences became a global leader in the open science revolution": I was thinking that mathematics has been leading this 'revolution'. Is it necessary to introduce such competition between disciplines?

2- In their analysis of the missing/incorrect annotation of data sets, I encourage the authors to discuss consequences on reproducibility as well. I understand that the text focuses on the FAIR principles, but considerations in terms of reproducibility are important.

3- I was surprised to find that the variables included in Figure 1 are mainly technical, and very little is provided on the sample or experimental design itself. Are samples linked to files? Is the biological condition annotated? These considerations are much more important for reuse than technical parameters that, as the authors state, can be roughly estimated from the data or instrumentation.

4- In figure 1, it is not clear what NA refers to, and how it differs from missing.

5- 'ontologies is repeated page 5

6- It is not clear how the curated projects were selected. A flow diagram showing how many

publications were considered, how many failed selection criteria, how many remained, would be most useful. The authors can look in figures displaying patient selection for clinical trials for inspiration.

7- A table 1 with descriptive statistics on the publications inspected would be useful: scientific/proteomic subfield, published or not, open access or not, number of samples, type of samples/workflow, etc.

8- It is not clear whether the restriction to a single factor is a limitation of SDRF or lessDRF, please clarify the text. This seems like a strong limitation for the encoding of biological/medical relevance, which I hope can be solved.

8- Discussion: 'realizes' should be 'realized', or rephrased

9- When discussing possible solutions to improve metadata annotation, the authors do not mention crowd-sourcing of information. It seems most relevant to enable the broader community to annotate public data sets and report discrepancies. For example, did the curation conducted by the authors result in new annotation for the projects? Were the errors detected corrected? If not, this is a missed opportunity.

10- Please clarify the definition of a project, and how it relates to scientific publications. There seems to be duplicates in Supplementary Table 2.

11- Please discuss the possibility to port lessDRF to other environments. It would be most useful to have it as part of the pride/ProteomeXchange submission system.

12- Manual: 'inhoud' should be translated.

Reviewer #3 (Remarks to the Author):

This manuscript introduces lesSDRF, an ontology-based Streamlit application that guides users through the annotation process using the Sample and Data Relationship Format (SDRF). Accurate, comprehensive, and standardized metadata annotation is essential to allow effective reuse of public data. However, despite the existence of the SDRF as a standardized metadata annotation format for proteomics studies, it is rarely used in proteomics publications and deposited data in proteomics data repositories, due largely to the effort required to prepare such annotations. Therefore, lesSDRF addresses a critical need in the proteomics field. The manuscript is clearly written, and a manual is provided as a supplementary file. My main suggest is that the manual is not very easy to follow. I would suggest that the authors select several representative studies from the 241 data sets and use them as examples to illustrate how to use lesSDRF, either through intermediate SDRF, or start with a new SDRF file. It would also be very useful to illustrate how to use this in large consortium projects (e.g., CPTAC) as these will be major data contributors.

One minor question is the initial selection in the web application. If I have a human cell line study, should I pick human or cell line? These should not be exclusive. Moreover, what will happen if I have a TMT plex with both tissue and cell line samples?

Reviewers comments:

Reviewer #1 (Remarks to the Author):

General comments

This manuscript introduces the work on developing a web-based tool for the Sample and Data Relationship Format (SDRF) file generation. The items of SDRF are collected from the GUI input by the user in a structured way. The author performed a baseline investigation to illustrate the necessity of SDRF file attachment in the submission of public data. The main text of the manuscript was used to introduce the significant meaning of publicly available metadata and the design of lesSDRF, but less attention was paid to the metadata contained and how the metadata can affect the reuse of the data. lesSDRF, as only an orphan web-based tool, may be useful to some academic cases, but we can also find some different tools on the PRIDE official website which can be used in the workflow of data submission. It is more convenient to use the official tools. For another consideration, we cannot figure out that lesSDRF can update automatically as the SDRF metadata standard updating from HUPO PSI. The metadata standard and the knowledge base of lesSDRF were not proven to be authoritative, comprehensive, or useful in practice.

We thank the reviewer for his/her comments and insights on the manuscript. In response to the point about lesSDRF as an 'orphan web-based tool' relative to existing tools on the PRIDE website, it should be noted that lesSDRF is not an independent project. It has been developed in close collaboration with the PRIDE team and is scheduled for integration into the PRIDE submission pipeline. This collaboration ensures that the tool is aligned with PRIDE's existing resources. Separately, lesSDRF also enjoys the backing of the HUPO-PSI initiative. As part of the HUPO-PSI team, we are closely aligned with developments in the SDRF (short for SDRF-Proteomics) data standard, allowing for a direct conduit between updates in the SDRF format and the lesSDRF tool. This ensures that the tool remains compliant with evolving metadata standards.

Special Concerns

In the Text mining, the 241 projects included in the statistics were published before 2018, the metadata may not be so concerned in those early submitted projects, these projects submitted recently may be better after the publication from Dai, C. et al. (Dai C et al. A proteomics sample metadata representation for multi-omics integration and big data analysis. Nat Commun. 2021 Oct 6;12(1):5854.). As we know, the number of submitted datasets to PRIDE has increased quickly in recent years.

We acknowledge the reviewer's observation regarding the temporal scope of the 241 projects included in our statistics. The reviewer rightly points out the impact of the 2021 Nature Communications publication of Dai et al. While we acknowledge that metadata concerns might have been less stringent in earlier projects, it is important to emphasize that the issue of incomplete or missing metadata extends beyond the repositories to the actual research articles themselves. Regardless of the era, reproducibility remains a cornerstone of scientific integrity, and it is compromised when metadata is lacking. Furthermore, as already mentioned in our original manuscript "Since SDRF's introduction as a supported file format in mid-2022, only 156 out of the 9,671 submitted datasets came with submitter-supplied SDRF annotation, bringing the total percentage of SDRF annotated projects in the entire PRIDE database to 3.9%." . These statistics quite clearly demonstrate that even since the formal introduction of the SDRF standard, there remains a considerable gap in user compliance for structured metadata submission to repositories like PRIDE.

In summary, while the publication of Dai et al. had an impact on the evolving awareness of the importance of metadata, the actual practice of submitting structured metadata is lagging. This suggests that, despite shifts in perception, the absence of an easy-to-use tool for metadata

annotation continues to result in missed opportunities for data integration, reusability and reproducibility, and thus in turn illustrates the importance of our lesSDRF tool.

Ref 23 and 25, why not use a journal-published citation?

Dai C, et al. A proteomics sample metadata representation for multi-omics integration and big data analysis. *Nat Commun.* 2021 Oct 6;12(1):5854.

Bouwmeester R, Gabriels R, Hulstaert N, Martens L, Degroev S. DeepLC can predict retention times for peptides that carry as-yet unseen modifications. *Nat Methods.* 2021 Nov;18(11):1363-1369.

We thank the reviewer for noticing this oversight and adapted our references accordingly.

Reviewer #2 (Remarks to the Author):

Claeys and colleagues propose an analysis of the annotation of publicly available proteomic data sets. As rightfully stated by the authors, the importance of metadata cannot be understated. They identify a substantial share of missing or incorrect annotation impairing the reuse of these data. To address this issue, the authors propose a web application that allows the generation of standardised metadata annotation. A much welcomed feature is the ability to load and extend partial files. As far as I tested, the tool works as wanted, providing much needed guidance on what the different fields and ontologies refer to. The code is publicly available and a manual is provided along with the tool. Whilst I support the publication of this work, I would appreciate it if the authors could consider the minor suggestions listed in order of appearance in the text.

We thank the reviewer for his/her kind appraisal of our work!

1- "the life sciences became a global leader in the open science revolution": I was thinking that mathematics has been leading this 'revolution'. Is it necessary to introduce such competition between disciplines?

We acknowledge the comment of the researcher. We did not want to induce competition between disciplines (hence also the 'a global leader' rather than 'the global leader'). To avoid any possible issue, we changed this sentence to read "The life sciences have already clearly highlighted the potential impact of open science" in the revised manuscript.

2- In their analysis of the missing/incorrect annotation of data sets, I encourage the authors to discuss consequences on reproducibility as well. I understand that the text focuses on the FAIR principles, but considerations in terms of reproducibility are important.

We thank the reviewer for this useful suggestion. To address this, we incorporated the following into the revised version of the manuscript:

"The resulting widespread access to publicly available proteomics data has not only allowed researchers to reuse, (re-)analyze, (re-)interpret, and integrate raw data from various experiments, yielding new insights and optimizing data potential, but it also plays a pivotal role in enhancing the reproducibility of research findings."

And:

“Clearly, this not only impedes maximal value extraction from public data, especially in large-scale data reuse, but it also poses significant challenges to the reproducibility of research.

3- I was surprised to find that the variables included in Figure 1 are mainly technical, and very little is provided on the sample or experimental design itself. Are samples linked to files? Is the biological condition annotated? These considerations are much more important for reuse than technical parameters that, as the authors state, can be roughly estimated from the data or instrumentation.

We appreciate the reviewer's keen observation regarding the focus on technical variables in Figure 1 and the lack of information on sample or experimental design. The text mining effort was conducted prior to the formal publication of the SDRF format, which explains why the annotation did not align with SDRF standards and why not all relevant parameters were identified. Additionally, the scope of the text mining was specifically targeted at metadata for further in-house reprocessing, and thus focused foremost on the technical information. However, from our experience, it is fair to state that projects that lack this basic (and, in fact, more easily documented) metadata, are also found lacking in more high-level (and therefore more complex to annotate) metadata such as experimental design.

We hope this clarifies the limitations and scope of the text mining and addresses the reviewer's concerns adequately.

4- In figure 1, it is not clear what NA refers to, and how it differs from missing.

NA means that no specific placeholder was provided for this element in the structured metadata in PRIDE. This has now been further clarified in the figure description “Figure 1. Comparison of metadata incompleteness in research articles and the corresponding PRIDE annotations. NA indicates that a specific placeholder for this metadata element was not available in the PRIDE structured metadata.”

5- 'ontologies is repeated page 5

This mistake was fixed.

6- It is not clear how the curated projects were selected. A flow diagram showing how many publications were considered, how many failed selection criteria, how many remained, would be most useful. The authors can look in figures displaying patient selection for clinical trials for inspiration.

The reviewer rightfully points to a possible lack of clarity concerning the selection process for curated projects. To briefly outline the selection process: Initially, 3,972 projects with human data published before 2020 were identified. From these, 355 projects were randomly selected for further consideration. Among these, one project had a publication still pending, making the paper inaccessible, while another had its paper retracted. Additionally, 91 projects were associated with closed-access papers. Further, we removed 21 cross-linking datasets as these did not align with the in-house purpose of our reprocessing effort. Ultimately, this resulted in a final set of 241 projects for

our study. As suggested, we've made a flow diagram showing this selection procedure and added it to the supplementary information.

7- A table 1 with descriptive statistics on the publications inspected would be useful: scientific/proteomic subfield, published or not, open access or not, number of samples, type of samples/workflow, etc.

We appreciate the reviewer's suggestion for a table with descriptive statistics on the inspected publications. As clarified earlier, all papers included in our study were both published and open access. Detailed information on each publication can be found in Supplementary Table 2 which contains comprehensive data.

We have also added a summary of labeling, experiment type, and sample type statistics to the supplementary information. However, it is worth noting that in-depth information on subfields and the number of samples was not annotated during our initial effort. Revisiting all the papers to gather this data is not feasible at this stage. We hope the currently available information adequately addresses the reviewer's concerns.

8- It is not clear whether the restriction to a single factor is a limitation of SDRF or lessDRF, please clarify the text. This seems like a strong limitation for the encoding of biological/medical relevance, which I hope can be solved.

We appreciate the reviewer's attention to the limitation caused by the restriction to a single factor in the manuscript. To clarify, this is not a limitation of the SDRF format, but was a design choice in lesSDRF. The restriction was initially implemented to minimize confusion around biological and technical replicates. However, the reviewer raises a valid concern about the potential impact of this limitation on encoding biological and medical relevance. In light of this feedback, we have meanwhile updated the lesSDRF code to remove this restriction. It is now possible to encode multiple factors. Thank you for bringing this important issue to our attention.

9- Discussion: 'realizes' should be 'realized', or rephrased

This mistake was fixed.

10- When discussing possible solutions to improve metadata annotation, the authors do not mention crowd-sourcing of information. It seems most relevant to enable the broader community to annotate public data sets and report discrepancies. For example, did the curation conducted by the authors result in new annotation for the projects? Were the errors detected corrected? If not, this is a missed opportunity.

We thank the reviewer for his/her insightful suggestion regarding the potential of crowd-sourcing for metadata annotation and error correction. Indeed, community involvement in annotating public datasets is a compelling strategy for enhancing data quality. As noted in the manuscript, "SDRF is actively used in large-scale initiatives on GitHub for post-hoc annotation of PRIDE datasets, with 220 projects annotated so far." This effort represents an ongoing community annotation initiative hosted on the GitHub repository of bigbio.

Regarding the text mining effort discussed in the manuscript, it was conducted prior to the formal publication of the SDRF standard. As such, the annotations generated through text mining are not in the SDRF format. However, since the development of lesSDRF, we have embarked on an in-house annotation effort, successfully annotating 15 projects that have been added to the GitHub repository mentioned above (PXD001324, PXD001325, PXD001326, PXD001524, PXD004540, PXD009254, PXD009261, PXD019423, PXD000228, PXD000440, PXD000605, PXD009737, PXD009752, PXD009754, and PXD009755). Additionally, collaborators within our group have already utilized lesSDRF for annotating their data prior to submission to PRIDE (PXD043476, PXD043300, and PXD043297).

We hope this clarifies the steps that the community, but also we ourselves, are taking to actively improve the quality of publicly available metadata.

11- Please clarify the definition of a project, and how it relates to scientific publications. There seems to be duplicates in Supplementary Table 2.

We thank the reviewer for seeking clarification on the definition of a project and its relationship to scientific publications, as well as for pointing out duplicates in Supplementary Table 2. A project refers to a dataset identified by a unique PXD identifier. It is worth noting that a single scientific publication can encompass multiple such projects, although the most usual scenario is still that one paper corresponds to one PRIDE project. The duplicates in the supplementary table are intentional and occur when a single PXD identifier encompasses multiple distinct types of data. This could range from different experiment types, such as label-free versus TMT-labelled data, to the use of different instruments for data acquisition. Splitting the project metadata over multiple rows prevents the mixing of metadata across such diverse experimental approaches in these cases.

12- Please discuss the possibility to port lessDRF to other environments. It would be most useful to have it as part of the pride/ProteomeXchange submission system.

We appreciate the reviewer's suggestion to consider the portability of lesSDRF to other environments, particularly its integration into the PRIDE/ProteomeXchange submission system. As mentioned in the manuscript, lesSDRF has been developed in close collaboration with the PRIDE team. Our ongoing partnership is directly focused on integrating lesSDRF in a modular fashion into PRIDE's existing pipeline and annotation efforts. Specifically, lesSDRF-generated SDRF files will feature a unique hash code that is recognizable by the PRIDE submission system, thereby streamlining the submission process for users.

Fully integrating lesSDRF into the PRIDE and ProteomeXchange submission systems will take some time to fully accomplish, and will involve sustained close coordination between the two different development teams. However, as we recognize the utility and efficiency that such integration would offer to the broader scientific community, we are exploring this possibility in more detail.

13- Manual: 'inhoud' should be translated.

This was corrected in the updated manual.

Reviewer #3 (Remarks to the Author):

This manuscript introduces lesSDRF, an ontology-based Streamlit application that guides users through the annotation process using the Sample and Data Relationship Format (SDRF). Accurate, comprehensive, and standardized metadata annotation is essential to allow effective reuse of public data. However, despite the existence of the SDRF as a standardized metadata annotation format for proteomics studies, it is rarely used in proteomics publications and deposited data in proteomics data repositories, due largely to the effort required to prepare such annotations. Therefore, lesSDRF addresses a critical need in the proteomics field. The manuscript is clearly written, and a manual is provided as a supplementary file.

My main suggest is that the manual is not very easy to follow. I would suggest that the authors select several representative studies from the 241 data sets and use them as examples to illustrate how to use lesSDRF, either through intermediate SDRF, or start with a new SDRF file. It would also be very useful to illustrate how to use this in large consortium projects (e.g., CPTAC) as these will be major data contributors.

We appreciate the reviewers insightful feedback on the manual. In response to the suggestions, we have taken steps to enhance the manual significantly. It is now more elaborate, delves deeper into the functionalities of lesSDRF, and presents information in a more fluid and coherent manner. Moreover, we made the decision to host the manual on GitHub. This ensures that it will be actively updated with new use-cases and mirror new developments in lesSDRF. As the application progresses, the manual will be continuously refined to ensure it remains a reliable and comprehensive resource for users.

One minor question is the initial selection in the web application. If I have a human cell line study, should I pick human or cell line? These should not be exclusive. Moreover, what will happen if I have a TMT plex with both tissue and cell line samples?

We acknowledge the reviewer's question regarding the initial selection in the web application for studies involving both human and cell line samples. This issue is a limitation rooted in the SDRF format and associated templates. For studies focusing on human cell lines, the 'cell line' option is recommended; this omits the need for human-specific parameters such as ancestry, sex, and age that are not applicable to a cell line. However, if a study includes both tissue and cell line samples, the 'human' selection is more appropriate as the tissue requires the inclusion of these human-specific parameters. It should also be noted that the selection of the cell line is carried out from the appropriate ontology (CL will implicitly contain information on its species of origin (e.g., human)). This is one of the reasons why the use of internally connected and well-constructed CVs is so critical to metadata application. Thus, while it may seem counterintuitive, the direct declaration of the cell line (in case a cell line is used), implicitly includes all relevant and available origin information of that cell line.

REVIEWERS' COMMENTS

Reviewer #2 (Remarks to the Author):

I thank the authors for the time and effort spent in considering my comments and congratulate them for this great piece of work. I have no doubt that it will be most helpful to the community.

Reviewer #3 (Remarks to the Author):

My concerns have been addressed. It is a good idea to host the manual on github.